# Epigenetic Biomarkers as Diagnostic Tools for Neurodegenerative Disorders

**DOI:** 10.3390/ijms23010013

**Published:** 2021-12-21

**Authors:** Olaia Martínez-Iglesias, Vinogran Naidoo, Natalia Cacabelos, Ramón Cacabelos

**Affiliations:** EuroEspes Biomedical Research Center, International Center of Neuroscience and Genomic Medicine, Bergondo, 15165 Corunna, Spain; neurociencias@euroespes.com (V.N.); serviciodocumentacion@euroespes.com (N.C.); rcacabelos@euroespes.com (R.C.)

**Keywords:** Alzheimer’s disease, Parkinson’s disease, DNA methylation, gene expression, sirtuin, diagnostic biomarker

## Abstract

Epigenetics is the study of heritable changes in gene expression that occur without alterations to the DNA sequence, linking the genome to its surroundings. The accumulation of epigenetic alterations over the lifespan may contribute to neurodegeneration. The aim of the present study was to identify epigenetic biomarkers for improving diagnostic efficacy in patients with neurodegenerative diseases. We analyzed global DNA methylation, chromatin remodeling/histone modifications, sirtuin (SIRT) expression and activity, and the expression of several important neurodegeneration-related genes. DNA methylation, SIRT expression and activity and neuregulin 1 (NRG1), microtubule-associated protein tau (MAPT) and brain-derived neurotrophic factor (BDNF) expression were reduced in buffy coat samples from patients with neurodegenerative disorders. Our data suggest that these epigenetic biomarkers may be useful in clinical practical for the diagnosis, surveillance, and prognosis of disease activity in patients with neurodegenerative diseases.

## 1. Introduction

Neurodegenerative disorders (NDDs) are major health issues in Western countries and are typically associated with aging. Alzheimer’s disease (AD) and Parkinson’s disease (PD) are the most common neurodegenerative disorders worldwide. AD is a progressive disorder that causes the irreversible loss of memory and cognitive function [1,2]. The main pathological features of AD are neuritic plaques and intracellular neurofibrillary tangles caused by the accumulation of amyloid-beta (Aβ) peptide and hyperphosphorylated microtubule-associated tau protein, respectively [3,4,5,6]. One of the most common causes of dementia in the elderly is vascular dementia (VaD), a syndrome caused by multi-focal vascular infarction and injury derived from cerebrovascular and cardiovascular disorders such as stroke and ischemic heart disease. Dementia affects 40–50 million people, with this number expected to rise to 145 million by 2050 [1]. PD, the second most common neurodegenerative disorder, affects 2% of the population over the age of 60 [7] and is multifactorial with genetic, environmental, cerebrovascular, and epigenetic components [4]. The motor dysfunction observed in clinical PD is due to the progressive loss of nigrostriatal dopaminergic neurons in the substantia nigra pars compacta and the formation and accumulation of Lewy bodies, as well as intracellular inclusions of α-synuclein [8]. 

Epigenetics is the study of reversible heritable changes in gene expression that occur without alterations to the DNA sequence, linking the genome and the environment [9,10,11]. The accumulation of various epigenetic alterations over the lifespan may contribute to neurodegenerative and cerebrovascular disorders [12,13,14]. DNA methylation, chromatin remodeling/histone modifications, and microRNA (miRNA) regulation are classic epigenetic mechanisms [9,11,14,15,16,17]. DNA methylation is a DNA methyltransferase (DNMT)-mediated reversible process in which methyl groups are added to cytosines in CpG nucleotides, converting them to 5-methylcytosines (5mC). This mechanism alters DNA stability and accessibility, regulating gene expression [18]. DNA methylation is usually a repressive mark [19], that attracts other silencing elements, such as methyl-CpG-binding proteins [20,21]. The addition of methyl groups is catalyzed by DNA methyltransferases (DNMTs) [22]. There are three DNMTs family proteins: DNMT1, DNMT2 and DNMT3 are all expressed in neurons [23] with different functions. DNMT1 maintains the methylation pattern after cell division and is responsible for the inheritance of methylation marks [24]. DNMT3a and DNMT3b are responsible for *de novo* methylation [25,26]. The ten-eleven translocations (TET) family of methyl cytosine dioxygenases (TET1, TET2, and TET3) oxidize and convert 5mC to 5-hydroxymethylcytosine (5hmC) [27]. 

Histone acetylation involves the transfer of an acetyl group to a lysine residue at the N-terminus of histones, decreasing positive charges within histones and weakening histone interactions with negatively charged DNA. This epigenetic modification promotes gene transcription by facilitating the binding of transcription factors and related enzymatic complexes to DNA [28]. Histone deacetylases, however, produce the opposite effect and inhibit gene expression [29]. Sirtuins (SIRTs) are nicotine adenine dinucleotide (NAD+)-dependent histone deacetylases (HDACs) that were first identified in yeast as transcriptional repressors; they are now known to occur in other species, including bacteria and eukaryotes [30]. In humans, the SIRT family comprises seven Class III histone deacetylases (SIRT1-SIRT7), each possessing different enzymatic activities, subcellular localizations, and physiological functions. All are involved in chromatin structure, cell cycle regulation, cell differentiation, cell stress response, metabolism, and aging [15,31]. SIRT1 is the most extensively studied among the mammalian SIRTs [29,32,33], and regulates neuronal differentiation, tumor progression, apoptosis, DNA stability, control gene expression, maintain chromosomal structure and control cell cycle progression [33]. SIRT1 is ubiquitously expressed in all tissues, including the brain [33]. This HDAC modulates neuronal differentiation, tumor and cell cycle progression, apoptosis, DNA stability, gene expression, and maintains chromosomal structure [20]; SIRT2 is involved in cell cycle regulation [15,20,32].

The identification of reliable biomarkers could aid early diagnosis of neurodegenerative diseases through the implementation of a personalized treatment program. There are currently no appropriate and reliable epigenetic biomarkers for the diagnosis, classification, or progression of NDDs [6]. Most current biomarkers rely on costly and/or invasive techniques such as neuroimaging or cerebrospinal fluid analysis [34]. A liquid biopsy presents a less expensive, but more comfortable, option. In recent years, several lines of research have focused on epigenetic biomarker-identification using more accessible fluids such as blood-derived samples; however, definitive epigenetic biomarkers for neurodegenerative and cerebrovascular diseases remain elusive. In the present study, we analyzed global DNA methylation, SIRT expression and activity, and expression levels of several genes with known important roles in NDDs, and identified novel epigenetic biomarkers for NDDs. These markers may be invaluable in clinical practice for the diagnosis and monitoring of neurodegenerative disease activity.

## 2. Results

### 2.1. Global DNA Methylation Is Reduced in Patients with Neurodegenerative Disorders

We previously showed that global DNA methylation (5mC) and global DNA hydroxymethylation (5hmC) levels are lower in buffy coat samples from patients with neurodegenerative disorders (AD and PD) and age-related cerebrovascular disease than in healthy patients [35]. Our aim in the present study was to examine whether global DNA methylation was modulated in buffy coat samples from a new, separate, patient cohort (*n* = 35) that further included individuals with Huntington’s disease and multiple sclerosis, in addition to those with AD, PD and PD-like disorders (Table 1). This study was inclusive and incorporated blood samples obtained from patients previously diagnosed with several types of neurodegenerative diseases (NDDs) and healthy (no-NDDs) subjects; NDD patients included those who exhibited dementia (NDD-D; AD and AD-like disorders such as vascular and mixed dementia), a range of parkinsonisms representing PD and PD-like disorders (NDD-PD), and other NDDs (NDD-O; Huntington’s disease, multiple sclerosis); the no-NDD group comprised 25 patients. Healthy individuals had 5mC levels of 3.58 ± 0.17%, but these values were significantly lower in patients with NDD (3.03 ± 0.19%; *p* < 0.05) (Figure 1A). 

Next, we asked whether there were differences in 5mC levels between patients within the NDD group. Patients with dementia (*n* = 21) had 5mC levels of 2.87 ± 0.27%, which were lower than in subjects with parkinsonism (*n* = 7) (3.03 ± 0.34%) and in patients with other types of NDDs (*n* = 7) (3.48 ± 0.27%); however, differences among these groups were not statistically significant (Figure 1B).

To further analyze the low 5mC levels observed in NDD patients, we used simple linear regression to evaluate the correlation between 5mC levels and age (Appendix A). We also analyzed the differences in 5mC levels between males and females (Appendix A). There was no correlation between 5mC levels and age (Appendix A), and neither were there differences between 5mC levels and gender (Appendix A) within the NDD and healthy patient groups (Appendix A); there was, furthermore, no correlation between 5mC levels and age in NDD + no-NDD patient samples (Appendix A), and no differences between genders. Since the ε4 allele of the apolipoprotein E (*APOE*) gene is the main genetic risk factor for late-onset AD [36], we further examined a possible link between global DNA methylation levels and the *APOE* gene variants *APOE 2.3*, *APOE 3.3*, *APOE 3.4*, and *APOE 4.4*; there was no correlation between 5mC levels and any of these APOE isoforms (Appendix A).

We previously showed that *DNMT3a* expression is downregulated in patients with various types of dementia, but not in patients with PD {35}. In the current study, we therefore decided to compare *DNMT3a* mRNA levels between patients in the NDD-D group only, and those subjects with no NDDs. *DNMT3a* expression in NDD patients with dementia decreased by 80% compared to healthy subjects (Figure 1C). We constructed receiver operating characteristic (ROC) curves to assess the value of reduced global DNA methylation (Figure 1D), SIRT activity and brain-derived neurotrophic factor (BDNF) expression in buffy coat samples as screening tests for NDDs. The area under the curve (AUC) for global 5mC levels was 0.66 (95% confidence interval (CI): 0.52–0.77, 40% specificity and 87.5% sensitivity, *p* = 0.0253).

### 2.2. SIRT Expression and Activity Are Reduced in Patients with Neurodegenerative Disorders

SIRTs are involved in various age-related signaling pathways, including responses to oxidative stress, mitochondrial dysfunction, protein aggregation, and inflammatory processes [37]. SIRT1 and SIRT2 may be protective against pathogenic processes in NDDs [38]. We therefore analyzed SIRT1 and SIRT2 expression in buffy coat samples from healthy subjects and patients with dementia (Table 1). SIRT1 mRNA levels were 75% lower in NDD-D patients than in individuals with no NDDs (*p* < 0.05) (Figure 2A); there were no significant differences in SIRT2 mRNA levels between these two groups (Figure 2B). To determine whether this reduction was specific to patients with dementia, we analyzed SIRT1 and SIRT2 expression in buffy coat samples from patients with parkinsonism. SIRT1 mRNA expression in NDD-PD patients decreased by 80% (*p* < 0.001) more than in patients without NDDs (Figure 2C). SIRT2 mRNA levels, however, were 20% lower than in the control group; this reduction was not statistically significant (Figure 2D). To discover whether these deacetylases were also altered in patients with NDD-D (Table 1), we then colorimeterically measured total SIRT enzyme activity in nuclear protein extracts from buffy coat samples. SIRT activity was completely absent in NDD-D patient samples compared to healthy subjects (*p* < 0.001) (Figure 2E). ROC curve analysis of SIRT activity as a potential biomarker of dementia revealed an AUC of 1.00 (95% CI 0.92–1.00, 100% specificity and 100% sensitivity, *p* < 0.0001) (Figure 2F). SIRT activity in NDD-PD patients was significantly lower than in subjects with no NDDs (*p* < 0.001) (Figure 2G). ROC curve data in NDD-PD patients showed an AUC for SIRT of 0.84 (95% CI 0.68–0.94, 81.25% specificity and 85% sensitivity, *p* < 0.0001) (Figure 2H).

### 2.3. Neurodegeneration-Related Gene Expression Is Altered in Neurodegenerative Disorders

Several genes have been linked to neurodegenerative disease pathogenesis [39,40,41,42,43]. We therefore next examined the expression of neurodegenerative-related genes in buffy coat samples in our patient cohorts (*n* = 9 healthy subjects; *n* = 6 NDD-D patients; *n* = 9 NDD-PD patients) (Table 1). Compared to healthy individuals, transcript levels of the pleiotrophic growth factor neuregulin 1 (NRG1) decreased by 50% in the NDD-D group (*p* < 0.05) (Figure 3A), and by 80% in NDD-PD patients (*p* < 0.01) (Figure 3B). Furthermore, mRNA expression of microtubule-associated protein tau (MAPT), associated with increased risk for AD and PD [44,45], was reduced by 65% in NDD-D patients (*p* < 0.001) (Figure 3C) but was five-fold higher in NDD-PD individuals (Figure 3D) than in subjects with no NDDs. BDNF is important in neuronal maintenance, survival, synaptic plasticity and the regulation of neurotransmission [46,47]. Our data showed a complete loss of BDNF expression in samples from the NDD-D group (*p* < 0.001) (Figure 3E) compared to healthy control subjects. ROC curve analysis of BDNF expression as a putative biomarker of dementia showed an AUC of 0.95 (95% CI 0.78–0.99, 100% specificity and 100% sensitivity, *p* < 0.0001) (Figure 3F). This reduction was profound, prompting us to examine BDNF expression in buffy coat samples from patients with parkinsonism (*n* = 5; Table 1). BDNF mRNA levels were 40-fold lower in NDD-PD patients (*p* < 0.001) (Figure 3G) than in healthy patients. ROC curve data analysis of BDNF expression from NDD-PD patients showed an AUC of 0.86 (95% CI 0.62–0.97, 100% specificity and 60% sensitivity, *p* < 0.0001) (Figure 3H).

## 3. Discussion

NDDs are multifactorial, complex, diseases in which genetic factors do not fully explain disease onset and progression. There is mounting evidence that environmental factors and epigenetics contribute to NDD pathogenesis. Environmental effects on gene expression, however, are mediated at least in part by various epigenetic mechanisms. DNA methylation is now recognized as a reliable biomarker in several diseases, including cancer, neurological disorders, and autoimmune disorders [48,49]. We previously showed that global DNA methylation is reduced in neurodegenerative and cerebrovascular diseases [35]. In the current study, we examined whether those findings could be replicated in a different, larger cohort of healthy and NDD subjects, that further included patients with other NDDs such as Huntington’s disease and multiple sclerosis.

The most common forms of senile dementia observed in the elderly are AD, VaD, and mixed dementia; together, they represent a continuum of pathologies with considerable overlap in terms of prevalence with age, symptomatology, etiology, risk factors, and comorbidity [50]. In this study, the NDD-D group, therefore, included patients with AD, VaD, and mixed dementia. In our previous study, we found no correlation between 5mC levels and psychometric parameters (Mini-Mental State Examination, MMSE) in healthy subjects nor in any patients in the NDD (PD, AD, and VaD) group; the only significant positive correlation was between age and 5mC levels in patients with PD (*p* = 0.0385) [35]. We did not collect Unified Parkinson’s Disease Rating Scale (UPDRS) data scores from patients in the present study. We did, however, use the MMSE for screening all patients for cognitive impairment. We found no correlation between 5mC, SIRT activity and BDNF mRNA levels versus MMSE values in healthy subjects nor in patients in the NDD group (data not shown). Consistent with our previous data, however [35], global 5mC levels decreased substantially more in the NDD group than in the non-NDD group; this reduction in global 5mC levels was also observed in other NDDs such as Huntington’s disease and multiple sclerosis. Among the different NDDs, 5mC levels were slightly lower in NDD-D patients than in the NDD-PD and NDD-O groups, but these differences were not statistically significant.

We previously detected a slightly significant (*p* = 0.0385) correlation between DNA methylation and age in patients with PD, but not in subjects with AD or VaD [35]. In the current study, however, there was no correlation between age and DNA methylation in healthy and patients with NDDs. Global DNA methylation changes with age, and lower 5mC levels are found in brain and blood samples from animal models and patients with NDDs [51,52,53,54,55,56,57,58,59,60]. In our previous study, the NDD (AD, PD, VaD) cohort included 101 patients; in terms of the age-range, 100% of NDD patients were older than 60 years old, with a positive correlation between age and 5mC levels found in PD patients only [35]. In the current study, the lower number of patients with NDDs (*n* = 35) and an age-range where 80% of NDD patients were older than 60 years old, led to no correlation between 5mC levels and age. In our study, since the majority of samples were obtained from individuals older than 60 years, patient age could therefore explain the disparity between our findings and those from other authors [54,55]. The expression of DNMTs also decreases with age [61]; DNMT1, DNMT3a, and DNMT3a2 levels are reduced in the frontal cortex and hippocampus in older human and mouse brains [62]. DNMTs are closely linked to memory and cognitive functions [63] and DNMT activity is required for the formation of associative memory and induction of long-term potentiation [64,65]. The loss of DNMT activity during certain periods of development significantly impacts cognitive function [66], suggesting that DNA methylation is important in regulating age-associated cognition. In the AD post mortem brain, DNMT1 expression and global 5mC and 5hmC levels are reduced within neurons in the entorhinal cortex layer II and hippocampus [57,58,62]. *DNMT3a* expression was furthermore reduced in buffy coat samples from patients with dementia [35,62]. In the present study, we confirmed that *DNMT3a* expression was lower in the NDD-D group than in healthy patients. Several studies, however, did not find significant differences between healthy and AD brain samples [60,62,67], or increased global DNA methylation levels in different regions of the brain in patients with AD [60]. These differences may be due to the analyses of different brain regions (e.g., whole brain, hippocampus, entorhinal cortex) in separate studies, or to the heterogeneity of pathological diagnoses in analyzed samples, since phenotypic heterogeneity in AD may influence DNA methylation levels. Those authors conducted experiments on serum samples but also on leukocyte samples; this sampling heterogeneity may further explain the variability in their data. The number of samples, in several cases, did not yield conclusive data [60,62,67]. Depending on the brain region, 5mC and 5hmC expression differ; both epigenetic marks are lower in astrocytes from patients with AD than healthy subjects [67,68]. In the brain of late-onset AD patients with Braak stage IV-VI pathology, obtained post mortem, there are no differences in 5mC or 5hmC levels in AD-resistant calretinin interneurons or microglia, nor any differences near β-amyloid plaque regions of interest, nor in plaque-free zones [67,68]. There were, however, high 5mC and 5hmC levels in neurofibrillary tangles.

DNA methylation-based age predictors are referred to as “epigenetic clocks” [69], with compelling evidence linking epigenetic age acceleration to common diseases [54]. In the present study, we did not find any correlation between 5mC levels and age in buffy coat samples from patients with NDDs, nor in subjects with no NDDs; the patient median age was 53 years, ranging from 20 to 86 years old. There was also no correlation between gender or APOE genotype and 5mC expression. However, global methylation levels increase in AD patients harboring the *APOE4* genotype [27,52]. The same authors report higher 5mC levels in whole blood from AD subjects, and a correlation between global methylation levels and psychometric parameters [27,52]. In those studies, global methylation was measured with a chemiluminescence substrate in whole blood samples, which contain different cell types with different methylation profiles [70]. In our study, however, we used an ELISA-like colorimetric assay to measure 5mC levels in buffy coat samples; this, along with the method of methylation quantification, could explain the disparities between the two studies.

Chromatin remodeling and histone post-translational modifications play important roles in NDDs. HDACs participate in transcriptional repression, leading to the generation of a compact chromatin structure. SIRT expression changes with aging and age-related NDDs [4,71,72]. Here, SIRTs promote lifespan and healthy aging by delaying the onset of neurodegenerative processes, and are new targets for treating neurodegenerative disorders [30,32,33,36]. Modulation of SIRT1 levels and/or activity is beneficial in various models of AD [36]; SIRT1 protects against β-amyloid plaque formation and ameliorates learning and memory deficits in animal models of AD [72]. SIRT1 deacetylates and reduces the levels of pathogenic p-tau proteins; SIRT1 silencing increases tau levels [36]. SIRT1 also regulates key PD-linked processes such as autophagy, apoptosis, mitochondrial dysfunction, oxidative stress and neuroinflammation [29]. Furthermore, SIRT1 overexpression blocks α-synuclein aggregation in in vivo and in vitro models of PD [73]. These findings are consistent with our present data, which show reduced SIRT1 expression in NDD-D and NDD-PD patients. SIRT2 is a highly conserved lysine deacetylase involved in aging, energy production and lifespan extension. SIRT2 levels increase with age and SIRT2 mediates processes involved in PD pathogenesis, including α-synuclein aggregation, microtubule dysfunction, oxidative stress, inflammation and autophagy [74]. High levels of SIRT2 are found in AD, PD and other neurodegenerative disorders, suggesting that it may therefore promote neurodegeneration [15]. SIRT2 may cause dopaminergic neuronal death [74]; in in vitro and in vivo models of PD, pharmacologic or genetic inhibition of SIRT2 protects against α-synuclein toxicity [36,75]. *SIRT2* variants influence biochemical, hematological, metabolic and cardiovascular phenotypes, and modestly affect pharmacoepigenetic outcome in AD [15]. However, SIRT2 may also be protective against neuronal injury [74]. In our study, we observed a small, non-significant reduction in *SIRT2* mRNA in samples from the NDD-D group; SIRT2 expression in patients from the NDD-PD group were unchanged. The present data suggest that SIRT1 expression is a better biomarker than SIRT2 for diagnosing patients with NDDs. Measurements of SIRT activity, however, consider global SIRT activity rather than just SIRT1 and SIRT2. Since reductions in SIRT activity were much higher than changes in SIRT1 expression, we cannot exclude the possibility that the expression of other SIRTs are also reduced in NDDs. To this, SIRT3 is implicated in the pathogenesis of AD, PD, amyotrophic lateral sclerosis, and Huntington’s disease [76]. SIRT3 mRNA and protein levels are reduced in the cerebral cortex of patients with AD and in the cortex of APP/PS1 double transgenic mice [76]. SIRT3-5 are active in mitochondria. In our study, we analyzed SIRT activity in nuclear protein extracts from buffy coat samples; given that SIRT3 is active in mitochondria, it may not be responsible for the decrease in SIRT activity. Furthermore, SIRT6 contributes to telomere maintenance, DNA repair, genome integrity, energy metabolism and inflammation, promotes longevity [77,78], regulates tau stability and phosphorylation [79], and is absent in patients with AD [78]. SIRT7, the least characterized SIRT, may be functionally significant in neural pathways and diseases [77]. Therefore, we cannot rule out the possibility that SIRT6 or SIRT7 may also be regulated in patients from the NDD-D and NDD-PD groups.

NRG1 signaling influences cognitive function and neuropathology in AD [80]. NRG1 attenuates deficits in spatial memory in AD transgenic mice in the Morris water-maze task and ameliorates neuropathology [80,81]. Concordantly, our data showed reduced *NRG1* mRNA levels in buffy coat samples obtained from patients diagnosed with various types of dementia. NRG1 further protects the mouse cerebellum against lipopolysaccharide-induced oxidative stress and neuroinflammation [43]. In samples from NDD-PD patients, our study showed that NRG1 expression decreased, similar to published data [42]; NRG1 is neuroprotective against 6-hydroxydopamine-induced toxicity in vivo [82]. MAPT expression is low in brain samples of patients with AD [43,44] and increases in PD [45]. Our data showed reduced MAPT expression in buffy coat samples of NDD-D patients.

Changes in the levels and activity of neurotrophic factors such as BDNF occur in several types of NDDs, including AD and PD [46,83,84,85,86,87,88]. BDNF levels are reduced in serum and brain samples from mouse models of tauopathy [83,84,85]. Intracerebroventricular administration of an adeno-associated virus carrying the gene encoding BDNF into mice produces stable BDNF expression, restores BDNF levels, prevents neuronal loss, alleviates synaptic degeneration, and attenuates behavioral deficits [89]. However, BDNF expression does not affect tau phosphorylation [84]. Reduced *BDNF* mRNA levels are found in the hippocampus and frontal cortex of patients with AD [84,85], suggesting that BDNF depletion or deficiency may contribute to the cognitive deficits in these patients. Low *BDNF* expression is also found in the plasma of patients with mild cognitive impairment (MCI) and AD [85,89]; the serum from patients with AD show significantly lower BDNF levels than those with MCI, confirming a connection between BDNF and AD; however, detection of BDNF is only possible in late stages of the disease [86]. Other types of dementia (frontotemporal, Lewy Body, or vascular) are associated with low BDNF levels, both in the systemic circulation and the central nervous system [83,84,85]. In our study, BDNF expression was almost non-existent in buffy coat samples obtained from patients suffering from various types of dementia, including AD and VaD.

Among individuals with PD, several pre-clinical and clinical studies report alterations in BDNF expression, implicating this neurotrophin in PD pathogenesis [87,89,90]. BDNF is crucial for dopaminergic neuron viability and maturation [82,87]. BDNF deficiency in the substantia nigra pars compacta is associated with the loss of dopamine-containing neurons, and patients with PD exhibit lower *BDNF* mRNA levels in the substantia nigra pars compacta than in healthy controls [83,89]. Neurons with low BDNF levels may be highly vulnerable to injury [88]. Inhibition of local BDNF production with an antisense oligonucleotide causes a significant loss of dopaminergic neurons in the rat substantia nigra pars compacta, showing that BDNF is important in neuronal survival [89,90]. In the present study, BDNF expression was dramatically reduced in buffy coat samples obtained from patients with PD. While pharmacologic treatment with levodopa increases BDNF expression [91], non-pharmacologic interventions such as cognitive rehabilitation speech therapy and physiotherapy may also positively affect BDNF levels [92].

With respect to existing diagnostic tools/algorithms for neurodegenerative diseases, while pathological analysis is regarded the gold standard in a wide range of disorders, it cannot be used to diagnose NDDs prior to the patient’s death. Other methods, such as positron emission tomography (PET) scanning or novel biomarkers (genomics and proteomics), may provide solutions and are being included into revised and improved diagnostic criteria [93]. The International Group of Alzheimer´s Precision Medicine Initiative, for example, was formed to assess the current state of the art for blood-based AD biomarkers. To date, 19 blood-based biomarkers have been chosen for further study towards the diagnosis of Alzheimer’s disease [93]. Recently, Huang Y et al. created Epigenome-Wide Association Studies (EWAS) plus, a computational technique that employs a supervised machine learning strategy, to expand the coverage of multiple EWASs to the entire genome rather than only about 2% of all CpG sites in the genome [94].

Blood DNA analysis is a non-invasive and inexpensive method for liquid biopsies, with diagnostic potential. Finding new non-invasive biomarkers for NDD diagnosis would be beneficial in treating these patients. Methylation levels in brain and blood samples from patients with PD are concordant [53]. In our current study, the area under the ROC curve (AUC) from the global DNA methylation assay in buffy coat samples from patients with NDDs was 0.66, ranking as “sufficient” [95]; the Youden index was 0.28 (40% specificity and 87.5% sensitivity). As higher AUC values correlate to better biomarker diagnostic strength, our data revealed that SIRT activity and *BDNF* expression are more reliable biomarkers; both had AUC values > 0.8, with a higher Youden index J. For patients with NDD-D, we calculated AUC values of 1.00 for SIRT activity (Youden index 1.00; 100% specificity, 100% sensitivity) and 0.95 for *BDNF* expression (Youden index 1.00; 100% specificity, 100% sensitivity). For patients with NDD-PD, we calculated AUC values of 0.84 for SIRT activity (Youden index 0.66; 81.25% specificity, 85% sensitivity) and 0.95 for *BDNF* expression (Youden index 0.66; 100% specificity and 60% sensitivity). Together, these findings show that global DNA methylation represents the biomarker with less diagnostic power than SIRT activity and *BDNF* expression for diagnosing and monitoring disease activity and treatment intervention in patients with dementia than in individuals with PD. Nonetheless, since DNA methylation levels, SIRT activity and *BDNF* expression all significantly decline in patients with dementia or PD, analyzing these three epibiomarkers may be useful in the diagnosis of NDDs. We propose combining the three markers to increase the efficacy of NDD diagnosis. Epigenetic modifications are reversible, and measuring DNA methylation levels, SIRT activity and BDNF expression may help clinicians monitor patient treatment responses.

## 4. Materials and Methods

### 4.1. Patients and Selection Criteria

Blood samples from patients with and without NDDs were obtained from the CIBE collection (C000925, 21 October 2013, EuroEspes Biomedical Research Center) after informed consent from all patients and/or legal caregivers. This collection follows standard ethical procedures according to Spanish law (*Organic Law on Biomedical Research*, 14 July 2007). The demographics and clinical characteristics of patients included in this study and stratified by severity (moderate, severe, and critical) and *APOE* genotype are shown in Table 1. Patients were diagnosed after undergoing the following tests: (i) clinical (neurologic, psychiatric) examination, (ii) blood and urine analyses, (iii) neuropsychological assessment (Mini-Mental State Examination (MMSE), Hamilton-A/D, Unified Parkinson’s Disease Rating Scale (UPDRS), Hoehn and Yahr Staging, Schwab), (iv) cardiovascular evaluation (EKG), (v) structural neuroimaging (brain MRI), (vi) functional neuroimaging (brain mapping), (vii) genetic assessment, and (viii) pharmacogenetic profiling including the study of several single nucleotide polymorphisms (SNPs) associated with PD, AD or VaD. Study procedures were reviewed and approved by the Institutional Review Boards of the International Center of Neuroscience and Genomic Medicine.

Blood samples (5 mL) from all patients were collected into anticoagulant-coated tubes and centrifuged at 2000× *g* for 15 min at 4 °C. The buffy coat fraction was then collected and stored at −40 °C until DNA or nuclear protein extractions. For RNA purification from blood samples, red blood cells were selectively lysed and centrifuged at 1500 rpm to precipitate lymphocytes. Lymphocytes were homogenized in Qiazol lysis reagent (Qiagen, Hilden, Germany) and samples stored at −40 °C.

### 4.2. DNA Extraction

DNA from peripheral blood lymphocytes was extracted using the QIAcube robotic workstation and QIAamp DNA Mini Kit (Qiagen), according to the manufacturer´s protocol. DNA purity and concentrations were measured with a microplate spectrophotometer (Epoch, BioTek Instruments, Winooski, VT, USA). Only DNA samples with 260/280 and 260/230 ratios above 1.8 were used in this study.

### 4.3. RNA Extraction

RNA from peripheral blood lymphocytes was extracted using the miRNeasy Mini Kit (Qiagen) as per the manufacturer´s protocol. Briefly, samples were incubated for 5 min at room temperature, mixed with chloroform, and then centrifuged at 12,000× *g* for 15 min at 4 °C to separate the organic and upper aqueous phases. RNA was extracted with the QIAcube, following the manufacturer´s instructions. RNA quality and concentrations were determined with a microplate reader (Epoch, BioTek Instruments, Winooski, VT, USA). Only RNA samples with 260/280 and 260/230 ratios above 1.8 were used.

### 4.4. Nuclear Protein Extraction

Nuclear protein extracts from peripheral blood lymphocytes were prepared using the EpiQuik Nuclear Extraction kit (Epigentek, New York, NY, USA) following the manufacturer´s specifications. Protein concentrations of nuclear extracts were determined with the Pierce bicinchoninic acid (BCA) Protein Assay (Life Technology, Rockford, IL, USA).

### 4.5. Quantification of Global DNA Methylation (5mC)

Global 5mC levels were measured colorimetrically using 50 ng DNA per sample with the MethylFlash Methylated DNA Quantification Kit (Epigentek, New York, NY, USA), according to the manufacturer´s instructions. Absorbance was measured at 450 nm with a microplate reader. To quantify the absolute amount of methylated DNA, we generated a standard curve using linear regression function (Microsoft Excel). The amount (ng) and percent of 5mC was then calculated with the formulae:5mC (ng) = (Sample OD − Blank OD)/(Slope × 2)
5mC (%) = 5mC (ng)/sample DNA (ng) × 100

### 4.6. Quantification of SIRT Activity

The SIRT activity study included healthy (*n* = 10), NDD-D (*n* = 12) and NDD-PD (*n* = 8) patients (Table 1). SIRT activity was measured by a colorimetric SIRT Activity/Inhibition kit (Epigentek, New York, NY, USA), as per the manufacturer’s instructions. Briefly, 50 ng nuclear protein extract was added to wells containing an acetylated histone-derived substrate and incubated for 90 min at 37 °C. The wells were washed and capture and detection antibodies added. The amount of deacetylated product, proportional to SIRT enzyme activity, was then measured by recording the absorbance at 450 nm in a microplate spectrophotometer.

### 4.7. Quantitative Real-Time RT-PCR

RNA was reverse-transcribed following the specifications of the High Capacity cDNA Reverse Transcription Kit (Applied Biosystems). Purified RNAs (200 ng) were copied into cDNAs using gene-specific primers under the following thermocycling conditions: 10 min at 25 °C, then 120 min at 37 °C, and 5 min at 85 °C.

Gene expression was quantified by qPCR using the StepOne Plus Real-Time PCR system (Applied Biosystems, Waltham, MA, USA). Each PCR reaction was performed in duplicate with the TaqMan Gene Expression Master Mix (Thermo Fisher, Waltham, MA, USA) and the specific TaqMan probes (Thermo Fisher) stated in Table 2. Results were then normalized to human GAPDH as an endogenous reference gene. Data analysis was performed using the comparative CT method with the StepOne Plus Real-Time PCR software, and presented as mean ± S.E.M.

### 4.8. Genotyping

SNPs and copy number variants (CNVs) were genotyped by qPCR amplification with TaqMan assays with the StepOne Plus Real-Time PCR system (Life Technology. Darmstadt, Germany) and TaqMan OpenArray DNA microchips for the QuantStudioTM 12K Flex Real-Time PCR System. Results were analyzed with Genotyper software (Thermo Fisher Scientific, Waltham, MA, USA).

### 4.9. Statistical Analysis

Statistical analyses were performed using GraphPad Prism (GraphPad Software, Inc., San Diego, CA, USA) and MedCalc version 16.4.3 (MedCalc Software). Data were tested for normality and equality of variances using the D’Agostino–Pearson Normality and Levene’s tests, respectively. Statistical significance was determined with a one-way ANOVA with post hoc Bonferroni correction for multiple comparisons, or unpaired *t* tests (GraphPad Prism, CA). The diagnostic accuracy of biomarkers was evaluated using non-parametric receiver operating characteristic (ROC) curves; the areas under the curve (AUC) were compared using the method of Delong et al. [96]. The Youden index (J = max {sensitivity + specificity − 1}) was used to identify the optimal (highest sensitivities and specificities) biomarker cut-off points. An AUC value of 0.5 shows a lack of diagnostic accuracy. The exact binomial method was used to estimate the 95% CIs of the AUC; AUC values are expressed with their 95% confidence intervals (CIs). Correlation analysis was performed using linear regression in GraphPad Prism. Data are presented as mean ± S.E.M.; * *p* < 0.05, ** *p* < 0.01 and *** *p* < 0.001 were considered statistically significant.

## Figures and Tables

**Figure 1 ijms-23-00013-f001:**
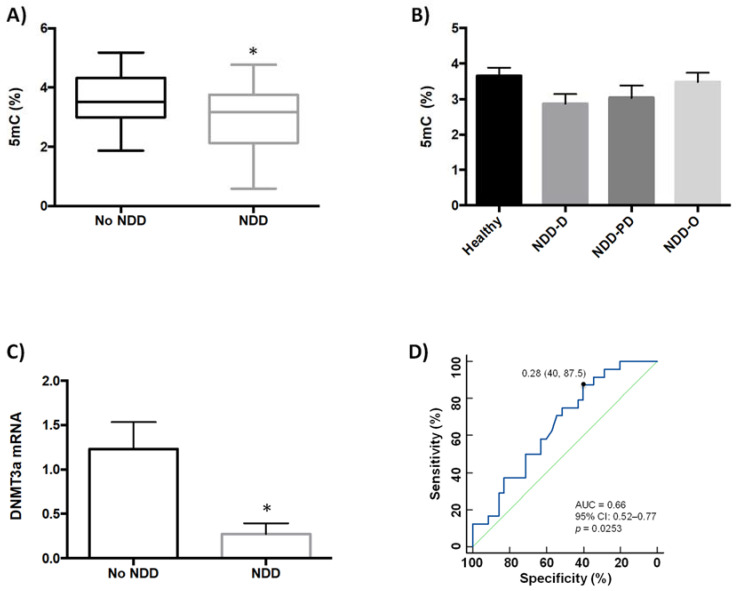
Analysis of global DNA methylation in blood samples from healthy subjects and patients with NDDs. (**A**) Global DNA methylation (5mC, %) levels were measured colorimetrically using buffy coat samples from healthy individuals (*n* = 2), and patients with NDDs (*n* = 35); unpaired *t* test (* *p* < 0.05). (**B**) 5mC levels (%) in healthy individuals (*n* = 25) and patients with NDD subtypes; NDD-D (*n* = 21), NDD-PD (*n* = 7), and NDD-O (*n* = 7); one-way ANOVA with post hoc Bonferroni correction for multiple comparisons (*p* = 0.256). (**C**) qPCR was performed in samples from the no-NDD and NDD-D groups with TaqMan probes for *DNMT3a*; unpaired *t* test (* *p* < 0.05). Data are presented as the mean ± S.E.M. (**D**) ROC curve to discriminate subjects without NDDs from patients with NDDs; AUC for 5mC was 0.66 (95% CI: 0.52–0.77), *p* = 0.0253. The optimal cutoff value according to the Youden’s index was 0.28 (specificity: 40%, sensitivity: 87.5%) (black filled circle). The diagonal green line is a reference line that corresponds to the ROC curve of a diagnostic test that has no diagnostic ability. 5mC, 5-methylcytosine; AUC, area under the curve; CI, confidence interval; DNMT3a, DNA methyltransferase 3a; NDD, patients with neurodegenerative diseases; no-NDD, individuals with no NDDs; NDD-D, NDD patients with dementia; NDD-PD, patients with a range of parkinsonisms representing PD and PD-like disorders; NDD-O; patients with other types of NDDs, such as Huntington’s disease, multiple sclerosis; PD, Parkinson’s disease; qPCR, quantitative real-time PCR; ROC, receiver operating characteristic.

**Figure 2 ijms-23-00013-f002:**
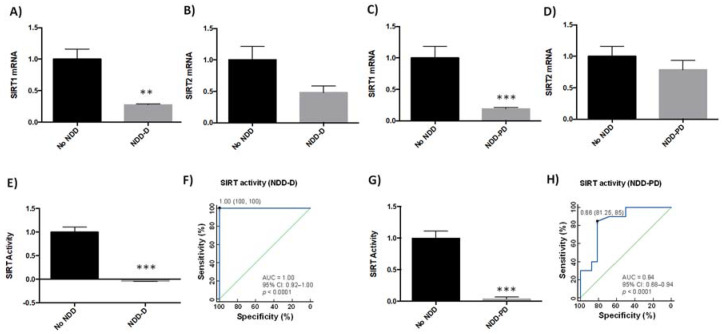
SIRT expression and activity in buffy coat samples from healthy individuals and patients with NDDs. qPCR was performed in samples from patients from the no-NDD (*n* = 9), NDD-D (*n* = 6) and NDD-PD (*n* = 9) groups with TaqMan probes for *SIRT1* (**A**,**C**) and *SIRT2* (**B**,**D**); unpaired *t* test (** *p* < 0.01; *** *p* < 0.001). SIRT activity was measured colorimetrically using buffy coat samples from healthy subjects (*n* = 15), patients with NDD-D (*n* = 16) (**E**), and in patients with NDD-PD (*n* = 19). (**F**) ROC curve analysis of SIRT activity; the AUC for SIRT in patients with dementia was 1 (95% CI: 0.92–1.00), *p* < 0.0001. The optimal cutoff value determined by the Youden’s index was 1.00 (specificity: 100%, sensitivity: 100%) (black filled circle). The diagonal green line is the ROC curve reference line. (**G**) SIRT activity levels in patients with parkinsonisms; unpaired *t* tests (*** *p* < 0.001). Data are presented as the mean ± S.E.M. (**H**) The AUC for SIRT from ROC curves for patients from the NDD-PD group was 0.84 (95% CI: 0.68–0.94), *p* < 0.0001. The optimal cutoff value according to the Youden’s index was 0.66 (specificity: 81.25%, sensitivity: 85%) (black filled circle). AUC, area under the curve; CI, confidence interval; no-NDD, individuals with no neurodegenerative diseases; NDD-D, NDD patients with dementia; NDD-PD, patients with parkinsonisms representing PD and PD-like disorders; PD, Parkinson’s disease; qPCR, quantitative real-time PCR; ROC 5, receiver operating characteristic 5; SIRT, sirtuin.

**Figure 3 ijms-23-00013-f003:**
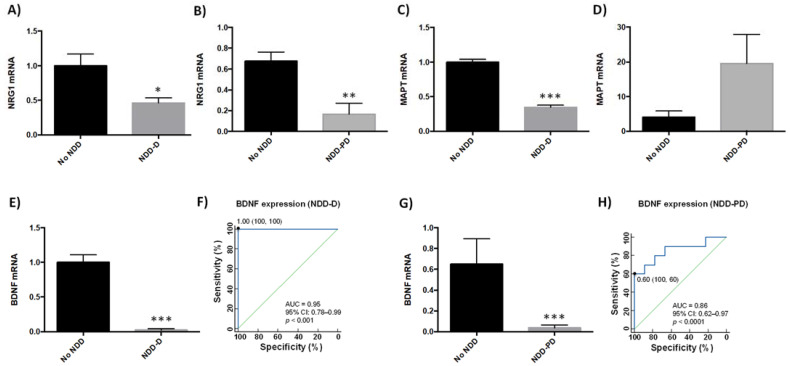
Gene expression in buffy coat samples from healthy individuals and patients with NDDs. qPCR was performed in samples from patients from the no-NDD (*n* = 9), NDD-D (*n* = 6) and NDD-PD (*n* = 9) groups with TaqMan probes for *NRG1* (**A**,**B**), *MAPT* (**C**,**D**) and BDNF (**E**,**G**); unpaired *t* tests (* *p* < 0.05; ** *p* < 0.01; *** *p* < 0.001). Data are presented as the mean ± S.E.M. (**F**) ROC curve, generated from *BDNF* mRNA expression data in patients with dementia, shows an AUC value of 0.95 (95% CI: 0.78–0.99), *p* < 0.0001. The optimal cutoff value determined by the Youden’s index was 1.00 (specificity: 100%, sensitivity: 100%) (black filled circle). The diagonal green line is the ROC curve reference line. (**H**) ROC curves for patients with parkinsonisms revealed an AUC for *BDNF* of 0.875 (95% CI: 0.62–0.97), *p* < 0.0001. The optimal cutoff value according to the Youden’s index was 0.60 (specificity: 100%, sensitivity: 60%) (black filled circle). AUC, area under the curve; BDNF, brain-derived neurotrophic factor; CI, confidence interval; MAPT, microtubule-associated protein tau; no-NDD, individuals with no neurodegenerative diseases; NDD-D, NDD patients with dementia; NDD-PD, patients with parkinsonisms representing PD and PD-like disorders; NRG1, neuregulin 1; PD, Parkinson’s disease; qPCR, quantitative real-time PCR; ROC 5, receiver operating characteristic 5.

**Table 1 ijms-23-00013-t001:** Study population demographics and classification of individuals under a range of neurodegenerative disorders.

			Total	Healthy	Dementia	PD	Others
**5mC study (Figure 1)**	Gender	N	60	25	21	7	7
		Age		47 ± 2.95	64 ± 3.34	70 ± 3.38	34 ± 8.13
		Male		11	6	4	5
		Female		14	15	3	2
	*APOE* Genotype	3.3		17	13	5	6
		3.4		5	4	1	
		4.4		2	3		
		2.3		1	1	1	1
		2.4					
**Sirtuin activity study (Figure 2)**	Gender
		N	50	15	16	19	
		Age		65.73 ± 2.23	65.44 ± 1.97	65.21 ± 2.07	
		male		6	6	9	
		Female		9	10	10	
	*APOE* Genotype	3.3		12	6	10	
		3.4		2	7	8	
		4.4			2		
		2.3		1	1	1	
		2.4					
**Gene expression study (Figure 1, Figure 2, Figure 3)**	Gender
		N	24	9	6	9	
		Age		67.71 ± 8.28	67.9 ± 6.1	79.89 ± 5.06	
		male		4	4	4	
		Female		5	2	5	
	*APOE* Genotype	3.3		5	3	2	
		3.4		2	1	5	
		4.4		1	1		
		2.3		1	1	1	
		2.4				1	

**Table 2 ijms-23-00013-t002:** TaqMan probes.

Gene	Reference
DNMT3A	Hs1027162_m1
SIRT1	Hs109006_m1
SIRT2	Hs1560239_m1
NRG1	Hs110158_m1
MAPT	Hs00902194_m1
BDNF	Hs329549_m1
GAPDH	Hs2786634_g1

## Data Availability

Not applicable.

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
