# Peer review of "Epigenetic Biomarkers as Diagnostic Tools for Neurodegenerative Disorders"

_ijms, 2021, doi:10.3390/ijms23010013_

Round 1

Reviewer 1 Report

The authors investigate the epigenetic markers associated with human neurodegenerative diseases and find alteration of status of DNA methlation and expression of some degeneration associated key proteins. These findings are interesting as they can contribute to disease biomarkers idenification which can add to diagnosis and disease monitoring.

However, some issues are raised which needs authors to clarify.

1, the authors has published a similar study in 2020 in the same journal, entitled "DNA Methylation in Neurodegenerative and Cerebrovascular Disorders". They also studied the DNA methylation status of neurodegenerative diseases with similar findings as in the current study. So there are overlaps between the previous report and the current study. They should cite and explain the situation.

2, In the current study, the authors have not found any correlations between age and DNA methylation status, which is different from previous study (reference 37, 52-61). They should explain the difference.

3, the authors only show RT-PCR results of some key proteins. However protein expression levels must be detected by Western blot analysis to get more solid conclusions, as mRNA level can not directly correlated to the exact level of expressed proteins.

4, the Sirt3 can be a key protein related to PD located in mitochondria (Oxidized nicotinamide adenine dinucleotide-dependent mitochondrial deacetylase sirtuin-3 as a potential therapeutic target of Parkinson’s disease. Ageing Research Reviews 62, 101107). Therefore the authors should check SIRT3 expression level and activity, which will be more impactful.

5, the neurodegenerative disease studied in the current study are too complicated including dementia, PD and others, which may be involved with complicated disease pathogenesis. One suggestion is to separate different diseases apart and study these epigenetic alterations, rather than combined different disease into one NDD group. This can be more reasonable and convincing. The authors can also investigate the disease progressive status and epigenetic status, for examples, whether the UPDRS score of PD patients can have any correlations with epigenetic changes.   

Author Response

We are grateful for the helpful and instructive criticism given and hope that you will find the revised manuscript “Epigenetic Biomarkers a Diagnostic Tools for Neurodegenerative Disorders” acceptable for publication as a Research Article in the special issue Genomics of Brain Disorders 3.0 in the International Journal of Molecular Sciences.

The points raised have been addressed as follows (reviewers’ comments are italicized); “Track Changes” has been used to reflect modifications to the revised manuscript.

Comment 1:

the authors has published a similar study in 2020 in the same journal, entitled "DNA Methylation in Neurodegenerative and Cerebrovascular Disorders". They also studied the DNA methylation status of neurodegenerative diseases with similar findings as in the current study. So there are overlaps between the previous report and the current study. They should cite and explain the situation.

Response:

Thank you to the reviewer. We have addressed the reviewer’s question in section “2.1 Global DNA methylation is reduced in patients with neurodegenerative disorders” on pages 2 and 3 (lines 92–107) of the revised manuscript. In the previous study (Martinez-Iglesias et al. 2020. DNA Methylation in Neurodegenerative and Cerebrovascular disorders. Int J Mol Sci. 21(6):2220), we compared global 5-methylcytosine (5mC), global 5-hydroxymethylcytosine (5hmC), and DNA methyltransferase (DNMT) expression in patients with Parkinson's disease (PD), Alzheimer's disease (AD), and vascular dementia (VaD), and healthy subjects. The results of that study suggested that analysis of global methylation status in patients with age-related neurodegenerative and cerebrovascular disorders may help improve diagnosis and patient follow-up. Given the limited number of studies that have measured DNA methylation levels in patients with neurodegenerative diseases, our aim in the current study was to therefore extend our earlier findings by examining whether global DNA methylation was modulated in buffy coat samples from a new, separate, patient cohort that also included individuals with Huntington's disease and multiple sclerosis, in addition to those with AD, VaD, PD and PD-like disorders. The overall objective was to clarify the possible use of 5mC (but also SIRT activity and BDNF expression) as a biomarker for those diseases, which may assist with early diagnosis and the implementation of a precision medicine program for treating individuals with neurodegenerative diseases.

Comment 2:

In the current study, the authors have not found any correlations between age and DNA methylation status, which is different from previous study (reference 37, 52-61). They should explain the difference.

Response:

In the revised manuscript, we have elaborated on the absence of any correlation between age and DNA methylation status in more detail in the Discussion section on page 8 (lines 264–276).

Comment 3:

the authors only show RT-PCR results of some key proteins. However protein expression levels must be detected by Western blot analysis to get more solid conclusions, as mRNA level can not directly correlated to the exact level of expressed proteins.

Response:

Thank you to the reviewer. Our aim in this study was to examine the mRNA levels of several important markers. We do not have any data on their protein expression at the current time as this was outside the aim and scope of our study. We will, furthermore, need to procure the corresponding antibodies, but given the limited timeframe for manuscript resubmission, it is not possible for us to presently conduct Western blot studies. We appreciate this comment, however, and agree that these are important experiments to perform. Our future objective is to investigate their protein expression by Western blotting and ELISAs.

Comment 4:

the Sirt3 can be a key protein related to PD located in mitochondria (Oxidized nicotinamide adenine dinucleotide-dependent mitochondrial deacetylase sirtuin-3 as a potential therapeutic target of Parkinson’s disease. Ageing Research Reviews 62, 101107). Therefore the authors should check SIRT3 expression level and activity, which will be more impactful.

Response:

We agree with and thank the reviewer for this insightful comment with respect to examining SIRT3 expression. We do not, unfortunately, have any data pertaining to this at the current time as we do not have TaqMan probes for qPCR to analyze SIRT3 expression. As indicated in this study, measurements of SIRT activity consider global SIRT activity rather than just SIRT1 and SIRT2 activities. It is, furthermore, worth noting that in our study we analyzed SIRT activity in nuclear protein extracts from buffy coat samples; given that SIRT3 is active in mitochondria, this suggests that it may not be responsible for the observed decreases in SIRT activity in the NDD-D and NDD-PD groups. Nonetheless, given that SIRT3 is the most predominant SIRT within mitochondria, quantifying SIRT3 expression will be an important future experiment to conduct.

Comment 5:

the neurodegenerative disease studied in the current study are too complicated including dementia, PD and others, which may be involved with complicated disease pathogenesis. One suggestion is to separate different diseases apart and study these epigenetic alterations, rather than combined different disease into one NDD group. This can be more reasonable and convincing. The authors can also investigate the disease progressive status and epigenetic status, for examples, whether the UPDRS score of PD patients can have any correlations with epigenetic changes.

Response:

In the revised manuscript, we have elaborated on the combination of the different neurodegenerative diseases in more detail on page 7 of the Discussion section (lines 241–250). In Figure 1A, we combined neurodegenerative diseases (NDDs) into a single group to provide an overall picture of the difference in 5mC levels between those patients and healthy subjects. In Figures 1B-D, 2 and 3, the NDD group was separated into patient sub-groups: a) NDD-D patients included those who exhibited dementia (AD and AD-like disorders such as vascular- and mixed dementia); b) NDD-PD were subjects exhibiting a range of parkinsonisms representing PD and PD-like disorders, and c) NDD-O patients with other types of NDDs, specifically Huntington’s disease and multiple sclerosis.

In our previous study (Martinez-Iglesias et al. 2020. DNA Methylation in Neurodegenerative and Cerebrovascular disorders. Int J Mol Sci. 21(6):2220), we found no correlation between 5mC levels and psychometric parameters (Mini-Mental State Examination, MMSE) in healthy subjects nor in any patients in the NDD (PD, AD, and VaD) group; the only significant positive correlation was between age and 5mC levels in patients with PD (p = 0.0385). We did not collect UPDRS data scores from patients in the present study. We did, however, use the MMSE for screening all patients for cognitive impairment. We found no correlation between 5mC-, SIRT activity- and BDNF mRNA levels versus MMSE values in healthy subjects nor in patients in the NDD group (see Figs. 1A-C, below). This information has been incorporated into the Discussion section on pages 7 and 8 (lines 246–258) of the revised manuscript.

Fig. 1 Correlation analysis between the psychometric parameter Mini-Mental State Examination (MMSE) and A) 5mC levels, B) SIRT activity and C) BDNF expression in patients with NDDs (n = 35).

We sincerely hope that we have addressed the comments to the satisfaction of the reviewer

Reviewer 2 Report

The authors recruited a cohort of neurodegenerative disease patients and generated a series of epigenetic datasets with routine statistical tests.

  1. The main idea of this study needs a clarification. The authors generated DNA methylation, chromatin remodeling/histone modification, and other epigenetic data for a small cohort of neurodegenerative diseases. Then the authors showed that these epigenetic data showed alternations between different diseases, supporting some statements in the literature. The data is not publicly available, and the analysis method is a routine operation. No quantitative diagnostic/prediction measurements (like accuracy) or novel biological insights were derived.
  2. The statistical significance is not very convincing, like p=0.0394 in the legend of Fig 1.
  3. The title claimed epigenetic biomarkers as diagnostic tools. Please describe how the existing diagnostic tools/algorithms perform on the neurodegenerative diseases.
  4. Words are abnormally broken, which may be caused by the formatting of another journal’s template, like “degenera-tive” and “expres-sion” in the Abstract.
  5. The page numbers need to be abbreviated consistently, e.g., Refs 65, 66, and 67.

Author Response

We are grateful for the helpful and instructive criticism given and hope that you will find the revised manuscript “Epigenetic Biomarkers a Diagnostic Tools for Neurodegenerative Disorders” acceptable for publication as a Research Article in the special issue Genomics of Brain Disorders 3.0 in the International Journal of Molecular Sciences.

The points raised have been addressed as follows (reviewers’ comments are italicized); “Track Changes” has been used to reflect modifications to the revised manuscript.

Comment 1:

The main idea of this study needs a clarification. The authors generated DNA methylation, chromatin remodeling/histone modification, and other epigenetic data for a small cohort of neurodegenerative diseases. Then the authors showed that these epigenetic data showed alternations between different diseases, supporting some statements in the literature. The data is not publicly available, and the analysis method is a routine operation. No quantitative diagnostic/prediction measurements (like accuracy) or novel biological insights were derived.

Response:

We appreciate this comment by the reviewer. The primary goal of this study was to assess the diagnostic utility of several biomarkers (global DNA methylation, sirtuin (SIRT) activity, and BDNF expression) for evaluating neurodegenerative diseases. The data presented here are based on buffy coat samples obtained from a small cohort of patients across a wide clinical and therapeutic spectrum; data evaluation included receiver operating characteristic (ROC) curve analysis. In the revised manuscript, we have now added sensitivity and specificity values for each of these putative biomarkers in the main text (5mC, last paragraph on page 3; SIRT activity, bottom of the paragraph on page 5; BDNF, bottom of the paragraph on page 6), figures (5mC, 1D; SIRT activity, 2F and H, and BDNF, 3F and H). ROC curves were plotted using MedCalc software (version 16.4.3; Ostend, Belgium). Maximum values of Youden’s index were used to select the optimal cut-off points yielding the highest sensitivities and specificities for each marker; these are indicated on the relevant graphs and their figure legends. For each ROC analysis, we have reported the area under the curve (AUC) and the associated p-value. The exact binomial method was applied to estimate the 95% confidence intervals (CIs) of the AUC. We have accordingly updated section “4.9. Statistical analysis” under Materials and Methods in the revised paper to describe this analysis (Page 13; lines 517–531).

The area under the ROC curve provides a number between 0 and 1, where values from 0.9–1 are considered excellent, 0.8–0.9 very good, 0.7–0.8 good, 0.6–0.7 sufficient. 0.5–0.6 bad and < 0.5 not useful. In our current study, the AUC value from the global DNA methylation assay was 0.66, ranking as “sufficient”; the Youden index was 0.28 (40% specificity and 87.5% sensitivity). As higher AUC values correlate to better biomarker diagnostic strength, our data revealed that SIRT activity and BDNF expression are more reliable biomarkers; both had AUC values > 0.8, with a higher Youden index J. For patients with NDD-D, we calculated AUC values of 1.00 for SIRT activity (Youden index 1.00; 100% specificity, 100% sensitivity) and 0.95 for BDNF expression (Youden index 1.00; 100% specificity, 100% sensitivity). For patients with NDD-PD, we calculated AUC values of 0.84 for SIRT activity (Youden index 0.66; 81.25% specificity, 85% sensitivity) and 0.95 for BDNF expression (Youden index 0.66; 100% specificity and 60% sensitivity). These findings show that SIRT activity and BDNF expression are better biomarkers than global DNA methylation and that these biomarkers may be better at diagnosing and monitoring disease activity and treatment intervention in patients with dementia than in individuals with PD. This information has now been added to the Discussion section on page 11 of the revised manuscript (lines 417–431).

Comment 2:

The statistical significance is not very convincing, like p=0.0394 in the legend of Fig 1

Response:

In the revised manuscript, for ROC curve data examination, we have now used MedCalc software (instead of GraphPad Prism) to generate graphs and for statistical analysis. In Figure 1 (ROC curve to discriminate 5mC levels in subjects without NDDs from patients with NDDs), p = 0.0253 and is statistically significant. However, we do agree that global DNA methylation represents the biomarker with the least diagnostic power from among the biomarkers we analyzed and we have added this point to Discussion section (lines 428-433).

Comment 3:

The title claimed epigenetic biomarkers as diagnostic tools. Please describe how the existing diagnostic tools/algorithms perform on the neurodegenerative diseases.

Response:

Thank you to the reviewer. This information has been added to the Discussion section on page 10 of the revised manuscript (lines 401–413). While pathological analysis is regarded the gold standard in a wide range of disorders, it cannot be used to diagnose NDDs prior to the patient's death. Other methods, such as PET scanning or novel biomarkers (genomics and proteomics), may provide solutions and are being included into revised and improved diagnostic criteria (Turner R Scott et al, 2020). The International Group of Alzheimer´s Precision Medicine Initiative, for example, was formed to assess the current state of the art for blood-based AD biomarkers. To date, 19 blood-based biomarkers have been chosen for further study towards the diagnosis of Alzheimer's disease (Turner et al 2020). Huang Y et al (2021) created Epigenome-Wide Association Studies (EWAS)plus, a computational technique that employs a supervised machine learning strategy, to expand the coverage of multiple EWASs to the entire genome rather than only about 2% of all CpG sites in the genome.

References

Turner, RS., Stubbs, T., Davies, DA., and Albensi, B. C. (2020). Potential New Approaches for Diagnosis of Alzheimer's Disease and Related Dementias. Frontiers in neurology, 11, 496.

Huang Y, Sun X, Jiang H, Yu S, et al. (2021). A machine learning approach to brain epigenetic analysis reveals kinases associated with Alzheimer's disease. Nat Commun., 12(1),4472.

Comment 4:

Words are abnormally broken, which may be caused by the formatting of another journal’s template, like “degenera-tive” and “expres-sion” in the Abstract.

Response:

The authors apologize for the error and thank the reviewer for pointing this out. This has been corrected in the Abstract.

Comment 5:

The page numbers need to be abbreviated consistently, e.g., Refs 65, 66, and 67.

Response:

The authors apologize for the error and have corrected this accordingly.

We sincerely hope that we have addressed the comments to the satisfaction of the reviewer

Round 2

Reviewer 1 Report

The revised manuscript can be published.